# Replicating shear-mediated self-assembly of spider silk through microfluidics

Jianming Chen[1,2,3,4], Arata Tsuchida [5], Ali D. Malay [1], Kousuke Tsuchiya[6], Hiroyasu Masunaga [7], Yui Tsuji[6], Mako Kuzumoto[6], Kenji Urayama[6], Hirofumi Shintaku [5] & Keiji Numata [1,6,8] ✉

The development of artificial spider silk with properties similar to native silk has been a challenging task in materials science. In this study, we use a microfluidic device to create continuous fibers based on recombinant MaSp2 spidroin. The strategy incorporates ion-induced liquid-liquid phase separation, pH-driven fibrillation, and shear-dependent induction of β-sheet formation. We find that a threshold shear stress of approximately 72 Pa is required for fiber formation, and that β-sheet formation is dependent on the presence of polyalanine blocks in the repetitive sequence. The MaSp2 fiber formed has a β-sheet content (29.2%) comparable to that of native dragline with a shear stress requirement of 111 Pa. Interestingly, the polyalanine blocks have limited influence on the occurrence of liquid-liquid phase separation and hierarchical structure. These results offer insights into the shear-induced crystallization and sequence-structure relationship of spider silk and have significant implications for the rational design of artificially spun fibers.

Spider dragline silk is a protein-based biopolymer that embodies an unparalleled combination of strength and flexibility that has elicited widespread scientific interest. Various efforts have been made to produce synthetic dragline silk in the lab from the component spidroin protein precursors; however, replicating the mechanical properties of the natural fiber remains a formidable challenge[1]. This is perhaps unsurprising, given that the performance of dragline silk arises from its complex hierarchical substructure[2,3], and to achieve this the spider employs a sophisticated mechanism that is orchestrated via conformational changes in the modular spidroin domains responding to precisely timed chemical triggers, combined with the physical forces that are generated within the confined geometry of the silk glands' spinning ducts[4,5].

Classical methods of artificial silk spinning, generally involves conversion of non-native protein feedstock into fibrous material through harsh denaturing conditions (such as alcoholic coagulation baths in wet-spinning)[6]. Recently[7,8], however, there has been growing interest in applying biomimetic techniques to produce fibers with native-like structure and functionality, under ecologically benign conditions. Here, the focus is on replicating the subtle biochemical processes that synergistically enable silk fiber self-assembly in the natural system. Such events include liquid-liquid phase separation (LLPS), considered a crucial step in the preassembly of diverse protein fibers[9,10], and which in spidroins is modulated through the repetitive domain (RPD) and C-terminal domain (CTD) in response to gradients of kosmotropic ions, such as phosphate[11,12]. An acidification gradient, meanwhile, mediates the rapid end-to-end multimerization of spidroin chains via the N-terminal domain (NTD), as well as an unfolding of the CTD[13,14]. Rheological effects such as shear and elongational flow are also major considerations as they facilitate the progressive alignment of spidroin chains inside the spinning ducts, and initiate the conversion of the extensive RPD from a largely disordered state toward

[1]Biomacromolecules Research Team, RIKEN Center for Sustainable Resource Science, 2-1 Hirosawa, Wako, Saitama 351-0198, Japan. [2]Research Institute for Intelligent Wearable Systems, The Hong Kong Polytechnic University, Kowloon, Hong Kong. [3]Research Centre of Textiles for Future Fashion, The Hong Kong Polytechnic University, Kowloon, Hong Kong. [4]School of Fashion and Textiles, The Hong Kong Polytechnic University, Kowloon, Hong Kong. [5]Cluster for Pioneering Research, RIKEN, 2-1 Hirosawa, Wako, Saitama 351-0198, Japan. [6]Department of Material Chemistry, Kyoto University, Nishikyo-ku, Kyoto 615-8510, Japan. [7]Japan Synchrotron Radiation Research Institute, 1-1-1, Kouto, Sayo-cho, Sayo-gun, Hyogo 679-5198, Japan. [8]Institute for Advanced Biosciences, Keio University, Tsuruoka, Yamagata 997-0017, Japan. ✉e-mail: keiji.numata@riken.jp

ordered intermolecular interactions that enable the formation of dispersed β-sheet nanocrystals that impart superior strength to the fiber[15,16].

Microfluidics present an exciting frontier in the assembly of structural proteins, wherein complex macroscopic architectures may be built, at least theoretically, through introduction of relevant physiological triggers in a flexible manner through customized channel arrangements and in the desired temporal sequence[17]. For artificial silk spinning, indeed, microfluidic-based methods have been recognized as holding the greatest potential for facilitating native-like fiber assembly, over more traditional techniques[18,19]. In particular, the confined dimensions and geometries of the spinning ducts, which determine critical flow parameters, can be approximated in the design of the miniaturized channels in the chip.

Previous studies have been reported that incorporate microfluidic principles in spinning artificial spider silk[20–25]. In terms of biomimetic fidelity, these run the gamut with a variety of recombinant spidroin sequences, microfluidic chip design, chemical trigger composition, and externalization of key processing steps being applied. A notable example showed fiber formation using three-domain ADF3 spidroin through sequential introduction of phosphate ions and pH decrease in a microfluidic device[22]. Crucially, however, no previous studies have reported the production of fibers with hierarchical nanoscale substructure, which is the hallmark of native silk self-assembly. A comparison of differences on the microfluidic spinning of recombinant spidroins was summarized in Supplementary Table 1. Interestingly, for spinning of regenerated silk fibroin (RSF), the forces generated inside the microfluidic chip have been quantified to show the distribution of shear and elongation rate along the channel, revealing the relationship between these forces and β-sheet contents[26,27]. However, in the case of recombinant spidroins, such quantitative analysis of the forces within

microfluidic channels have not been carried out in rigorous detail to link the forces with the fiber formation and change of conformation. In comparison with RSF, recombinant spidroins in the native state are more responsive to naturally occurring triggers leading to self-assembly behavior. Therefore, to quantify the forces, such as shear stress, under native-like conditions is of great interest and significance for better understanding of spider silk self-assembly mechanism.

Here we present a rationally designed microfluidic system that aims to emulate the functionality of the natural silk spinning apparatus by integrating recent insights regarding the in vivo self-assembly process. By sequential administration of biomimetic chemical gradients that trigger phase separation and nanofibril formation, and through quantifiable shear effects, we demonstrate the complete in situ assembly of hierarchically structured silk fibers from recombinant MaSp2 precursors, with tunable β-sheet abundance and at near-instantaneous speeds.

## Results

### Design of biomimetic microfluidic device

We aimed to design a biomimetic microfluidic device (Supplementary Discussion) that approximates the native-like chemical and physical gradients within the channel for the investigation of spider silk fiber assembly. The architecture of the chip is shown in Fig. 1a. It includes 3 inlets, namely for the (1) precursor spidroin solution, (2) kosmotropic ions at neutral pH, and (3) low-pH fibrillization trigger. The architecture provides for two sequential mixing regions, followed by an extended channel terminating at an outlet, and defines three regions along the device where successive stages of self-assembly are monitored (sections A, B and C). The design is rationalized as follows: upon flow initiation, the streams originating from (1) and (2) merge, exposing the spidroin to a kosmotropic anion gradient similar to that found

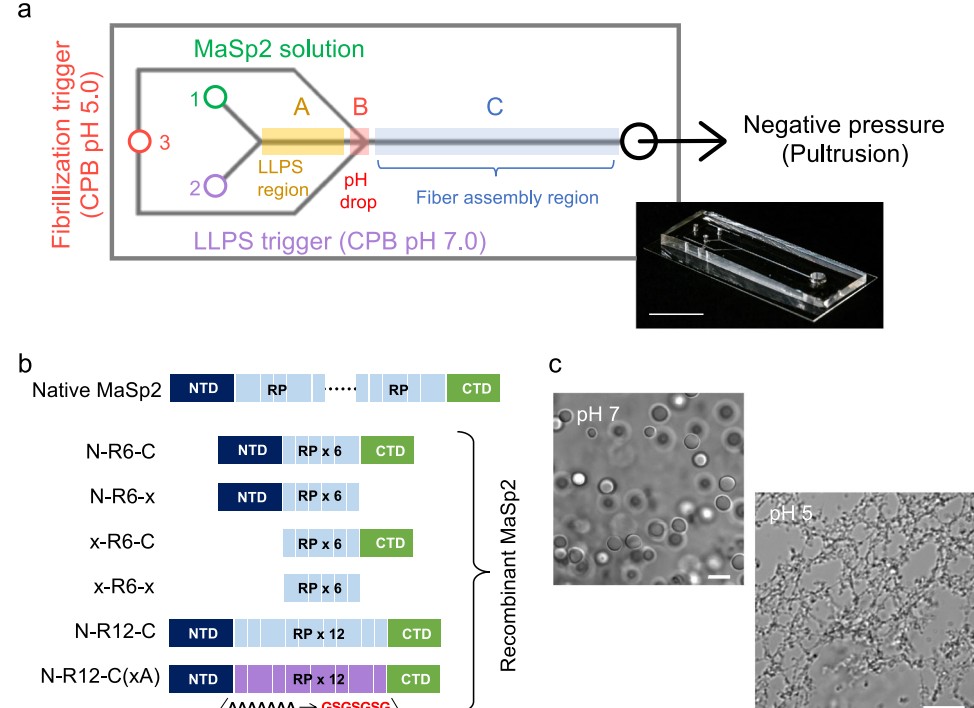

**Fig. 1 | Design of MaSp2 constructs for biomimetic spinning in a microfluidic device. a** Layout of the microfluidic device, consisting of three inlets and one outlet with a negative pressure controlling system. The spidroin was first mixed with 50 mM CPB at pH 7 and then subjected to 1 M citrate–phosphate buffer (CPB) (pH 5) at the intersection. The channel along which MaSp2 assembled was divided into sections A, B and C for comparison in terms of the morphology and secondary structure. The inset shows a picture of the actual microfluidic device. Scale bar,

1 cm. **b** The different recombinant MaSp2 constructs used in this study, representing various combinations of the N-terminal domain (NTD), repeat (RP) and C-terminal domain (CTD). For N-R12-C(xA), the poly-Ala repeat normally found in the repetitive region was replaced by GSGSGSG. **c** N-R12-C in the form of droplets and nanofibrils obtained by exposure to 0.5 M CPB at pH 7 and 5, respectively. Five independent experiments were performed with similar results. Scale bar, 5 μm.

in the spider spinning ducts, which is predicted to trigger LLPS of MaSp2, with the nascent protein droplets being carried downstream via laminar flow through section A. Subsequent contact with the stream originating from (3) along opposite directions creates a pH gradient, which mimics the natural acidification effect, and which is expected to trigger solidification of the spidroin condensates into fibrillar networks (section B). Section C constitutes a straight channel with dimensions of $2 \, cm \times 80 \, \mu m \times 62 \, \mu m$, where mechanical deformation is predicted to promote the assembly of extended silk-like fibers.

It is important to state that we chose a simplified design where each individual channel has a uniform cross-section, and an increase in width from section B ($60 \, \mu m$) to C ($80 \, \mu m$), in order to focus on the effects of shear in modulating fiber assembly and crystallization, while minimizing effects of extensional flow. This is contrast to the more complex (i.e., convergent) geometries found in the natural spinning ducts, where the progressive narrowing of the ducts toward the spinneret produces a strong extensional flow that contributes to the alignment of silk molecules. Significantly, the sample flow in our system is initiated through the application of negative pressure (vacuum) from the outlet, a feature not common in previous strategies for the assembly of silk-like fibers, where the flow was typically generated either via positive pressure[23] or syringe-pumping[28,29] techniques; this was implemented to mimic the observation that natural silk fibers are "pulled" from the organism (pultrusion) rather than being "pushed out"[14,30]; additionally this enabled fine-tuned control of the forces generated within the microfluidic device.

## Native-like silk assembly from LLPS to nanofibrillation to fiber formation

As spider silk building blocks, we used recombinant MaSp2 dragline spidroin bearing a range of domain architectures (Fig. 1b) including complete constructs of N-R6-C and N-R12-C (with 6 and 12 tandem repeats, respectively), the truncated constructs N-R6-x, x-R6-C, and x-R6-x, as well as the three-domain N-R12-C(xA) variant that substitutes the polyalanine blocks with alternating glycine and serine residues (GSGSGSG), to evaluate the sequence dependence of β-sheet formation during fiber assembly. The prepared spidroins were confirmed by SDS-PAGE (Supplementary Fig. 1).

Outside of the microfluidic system, three-domain MaSp2 can readily be induced to form LLPS droplets by exposure to kosmotropic ions at neutral pH (Fig. 1c), whereas under acidic conditions (pH 5.5–5.0), the combined effects of LLPS and NTD-mediated multimerization produced spontaneous and rapid network assembly (Fig. 1c). In this study, we used a hybrid citrate-phosphate buffer (CPB) system as the general kosmotropic trigger, in light of its improved buffering capacity throughout the relevant pH range (7.5–5.0) compared to the individual phosphate and citrate components.

The stepwise biomimetic process was integrated into the microfluidic system, and optimized for generating hierarchically-structured MaSp2 fibers with minimum intervention. The method itself simply involves sequential addition of components at inlets 3, 2, and 1 under constant negative pressure, followed by immediate cessation of the pump (Supplementary Movie 1). Conditions for effective fiber formation, which we define as the generation of a single continuous fiber inside the microfluidic channel, excluding fiber breakage, or formation of aggregates or their mixtures, were established empirically. Here, the composition of the fibrillization buffer was found to be critical, with optimal results obtained with CPB at a concentration between 1–1.5 M and pH value of 5.0, as well as the magnitude of the applied pressure (Supplementary Figs. 2, 3).

Figure 2a provides an overview of the results obtained using labeled construct N-R12-C at 50 mg/ml (inlet 1), 50 mM CPB pH 7.0 (inlet 2), and 1.0 M CPB pH 5.0 (inlet 3), under −90 kPa pressure. Condensed MaSp2 structures could be observed throughout the

length of the channels, with a distinct evolution of the mesoscale morphology through successive stages of the device. Along section A, the channel wall on the side of inlet 1 was densely populated with roughly spherical or amorphous deposits of MaSp2 condensate, with partial droplet fusion and surface wetting properties consistent with LLPS formation and downstream migration of the protein droplets via laminar flow (Fig. 2a–c). This result was in agreement with the dynamic formation and fusion behavior of droplets as shown in Fig. 1c. The fact that MaSp2 was able to undergo extensive LLPS despite the relatively limited surface contact between the buffer phase and the kosmotropic trigger due to laminar flow indicates that the initial conditions were close to the phase separation boundary, which ensured prompt phase separation when the two streams came into contact.

In section B, the MaSp2 condensate transitioned into a flattened sheet morphology with a granular surface, an abrupt change triggered by the combined effects of the acidification gradient, which triggers fibrillization, and the shear effects imposed by the changes in channel geometry. In addition to the visible change of morphology in section B, the corresponding structure of spidroins were also affected by both chemical and physical triggers introduced in this fibrillization stage (see below). Further downstream, in section C, we see the formation of a continuous unbroken fiber throughout the length of the channel, with circular diameter of 5–10 μm. Here, the hierarchical substructure of the fiber is clearly evident, consisting of densely-packed nanofibrils oriented along the longitudinal axis (Fig. 2d, e; Supplementary Movie 2), validating the biomimetic self-assembly strategy. In effect, the formation of aligned nanofibril bundles in section C of the device can be considered as analogous to the assembly of the random protein network structures outside of the device upon mixing MaSp2 with CPB at pH 5 (Fig. 1c), with the additional factor of fibril alignment provided by the directional flow within the channels; in other words, 2D network structure can be transformed into 1D nanofibrils after uniaxial orientation of the protein network. After formation, the fibers could be recovered from the device for further analysis. Significantly, the fibers were able to retain structural integrity in pure water and demonstrate high flexibility (Fig. 2f). In some localized areas, the overall fiber architecture did break apart, which clearly revealed an underlying substructure consisting of individual protein fibrils with high aspect ratio and oriented along the longitudinal axis of the fiber Fig. 2g. A visualization of the entire length of a microfluidic device following fiber assembly of construct N-R12-C is presented in Supplementary Movie 3, which provides further evidence of hierarchical organization.

Interestingly, whereas the complete constructs (i.e., N-R6-C and N-R12-C) produced fibers with microscopically indistinguishable morphologies, the truncated variants N-R6-x, x-R6-C, and x-R6-x failed to produce any fibers under the same conditions, underscoring the crucial interplay between the different domains for successful self-assembly of spider silk under native conditions[12]. In addition, while the three-domain N-R12-C(xA) lacking polyalanine blocks displayed similar morphology to constructs of N-R6-C and N-R12-C in section A and B, the fiber structures produced in section C were often ruptured inside the channel, reflecting the apparent mechanical weakness of these structures (Supplementary Movie 4).

## Quantification of shear-induced crystallization

While the above demonstrates a microfluidic platform capable of inducing rapid self-assembly of recombinant MaSp2 into hierarchically structured fibers in response to sequential physiological triggers, another major concern is whether such a system can successfully induce the formation of β-sheet structures, which are the main source of tensile strength in spider silk.

Changes in protein conformation within the microfluidic chip were monitored by confocal Raman spectroscopy, with $0.8 \, cm^{-1}$ resolution (Fig. 3a). Spectra collected from native spider dragline silk

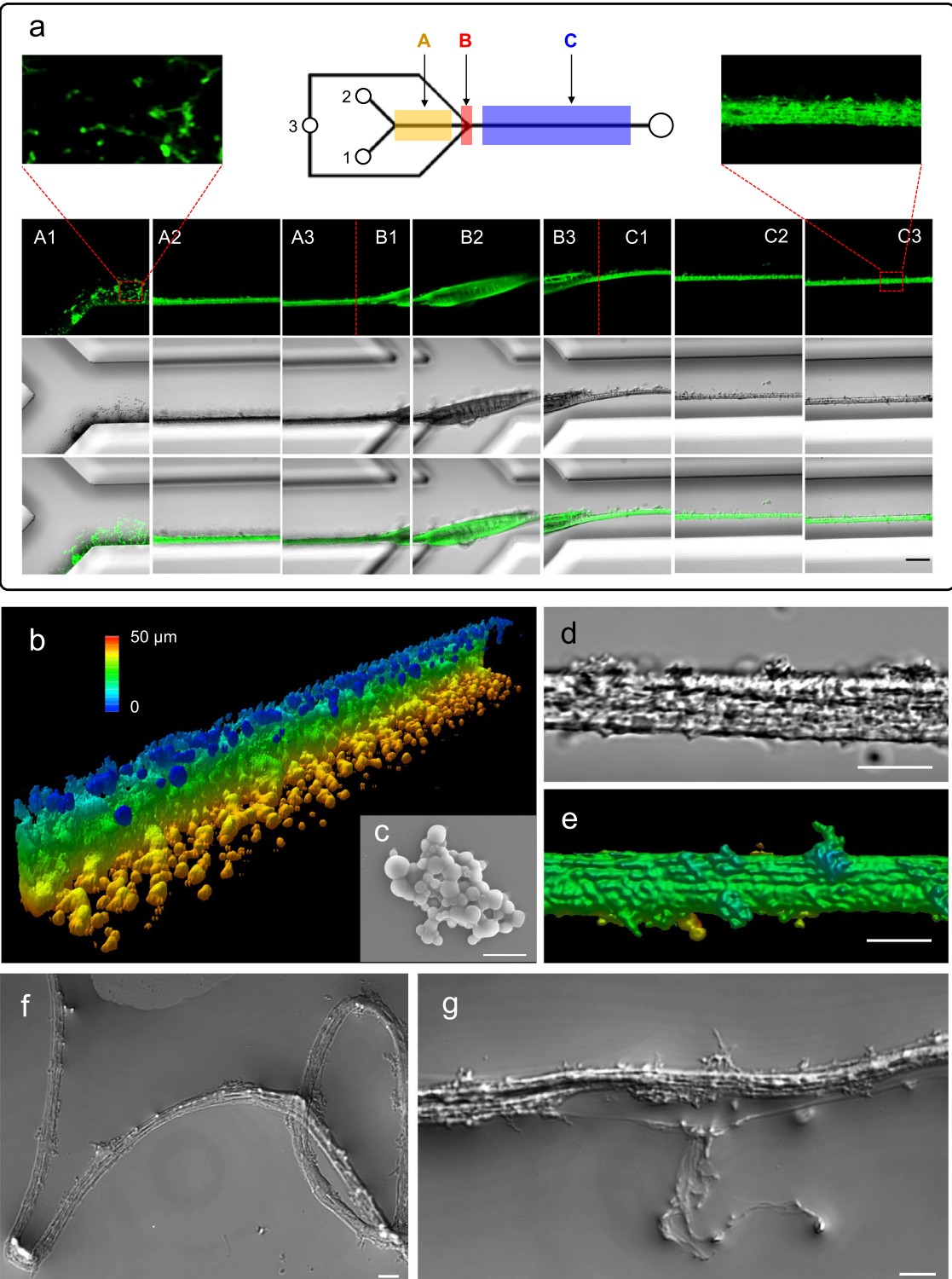

**Fig. 2 | Morphology of MaSp2 droplets, nanofibrils and fibers as characterized by CLSM and SEM. a** N-R12-C labeled with DyLight-488 was transferred to the microfluidic device for spinning under native-like conditions. Three independent experiments were performed with similar results. Scale bar, 30 μm. **b** 3D reconstruction of N-R12-C droplets distributed along the channel. Scale bar, 5 μm. **c** Surface structure of N-R12-C droplets induced by LLPS inside the microfluidic device as visualized by SEM. Two independent experiments were performed with similar results. Scale bar, 10 μm. **d** N-R12-C fiber assembled inside the channel in section C3 with a hierarchical organization. Three independent experiments were performed with similar results. Scale bar, 10 μm. **e** 3D reconstruction of N-R12-C fiber aligned along the channel in section C3. Two independent experiments were performed. Scale bar, 5 μm. **f** Oriented nanofibrils observed from the flexible continuous fiber in Milli-Q water after recovering it from the microfluidic chip. Three independent experiments were performed with similar results. Scale bar, 10 μm. **g** Nanofibrils split from the fiber to show hierarchical organization upon exposure to the Milli-Q water. Three independent experiments were performed with similar results. Scale bar, 10 μm.

fiber (Supplementary Fig. 4), used as a reference, produced well-defined peaks at the expected locations of 1670 (amide I, C = O stretching), 1615 (Tyr), 1452 (CH₃ asymmetric bend, CH₂ bending), 1242 (amide III, N−H bend + C−N stretching), 1175 (Tyr), and ~1078/1094 cm⁻¹ (skeletal $C_\alpha$−$C_\beta$ stretching). Of these peaks, the amide I, amide III and skeletal C−C peaks are sensitive to changes in secondary structure, particularly with regard to β-sheet conformations[31]. Normally, the peaks within the amide I or III regions can be deconvoluted to calculate the secondary structure composition[32]. However, the PDMS and coverslip produced overlapping signals at 1266 and 1098 cm⁻¹, respectively, and thus we focused on the area surrounding the amide I region for the quantitative analysis of secondary structure content. Preliminary tests confirmed that outside of the microfluidics context, three-domain MaSp2 exposed to the biomimetic chemical triggers, but in the absence of mechanical deformation, failed to produce β-sheet conformations (Fig. 3b); while as positive control, MaSp2 condensate subjected to 90% methanol treatment produced the characteristic amide I peak for β-sheet conformation at 1668 cm⁻¹.

Raman measurements using N-R6-C, N-R12-C, and N-R12-C(xA) revealed an evolution of protein secondary structure (Fig. 3c) during self-assembly in the microfluidic device, complementing the observed changes in morphology. During the LLPS droplet regime (section A), all samples showed spectra consistent with largely disordered/α-helical conformations, with a peak at ~1654 cm⁻¹. Significantly, for N-R12-C, the progressive emergence of an additional peak at ~1664 cm⁻¹ was observed, attributed to β-sheet structures. Deconvolution of the amide I region from 1580 to 1720 cm⁻¹ provides estimates of the secondary structure contributions (Supplementary Fig. 5). For N-R12-C, as shown in Supplementary Table 2, the estimated β-sheet content increased from 12.5% in section A to 22.5% in section B and finally to a maximum of 29.2% in section C. In contrast, for N-R6-C, with half the number of tandem repeats, no obvious β-sheet peaks were detected, despite the assembly of morphologically similar fibers; we hypothesize a higher pressure threshold for triggering structural change in the 6- versus the 12-repeat construct, with an applied pressure of −90 kPa (the upper limit for our system) being evidently insufficient to drive abundant β-sheet formation in N-R6-C. Likewise, for the mutant N-R12-C(xA), a secondary structure transition was not observed, in line with poly-alanine forming the basis for the β-sheet crystalline component of the silk fibers.

It is worth noting that while the N-R12-C was exposed to slightly higher shear in section A compared to B (Fig. 4c, e), the more considerable increase in β-sheet content (Supplementary Table 2) was detected in the latter section. It can thus be deduced that simple combination of shear and LLPS initiated by ions exchange in section A is not enough to induce the conformational transition. It is thus apparent that the fibrillization induced by acidification in section B, along with the associated increase in viscosity, also serve as essential prerequisites in the synergistic process of β-sheet formation in our microfluidic system.

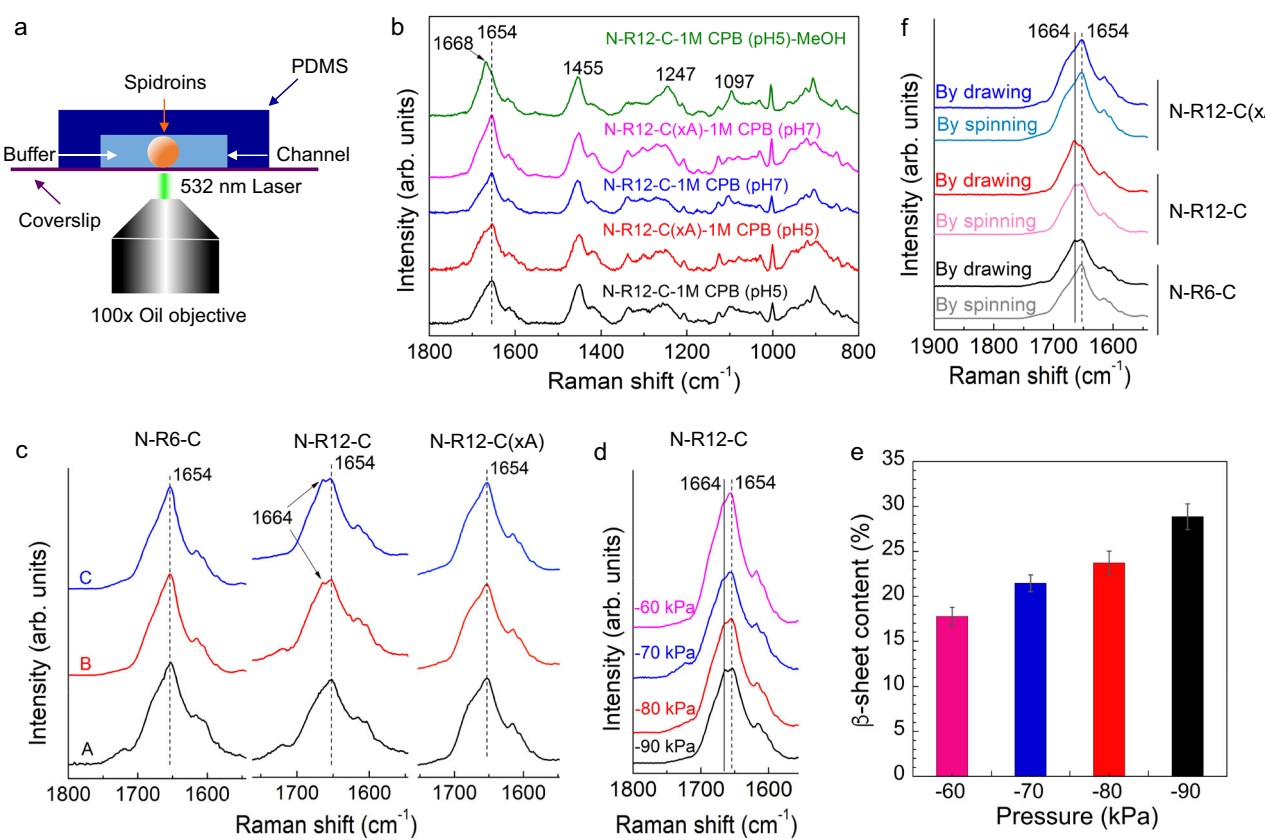

**Fig. 3 | Characterization of the β-sheet formation of MaSp2 fiber formed in the microfluidic device or by manual drawing. a** Secondary structure of the MaSp2 fiber determined by Raman spectroscopy immediately after fiber assembly inside the channel. **b** Negative control experiment of N-R12-C and N-R12-C(xA) outside the microfluidic device. N-R12-C and N-R12-C(xA) were gently mixed with 1 M CPB at pH 5 and 7 on a cover slide at a 1:1 volume ratio. A positive control was conducted by immersing N-R12-C nanofibrils in 90% methanol (MeOH) solution for 5 min after removal of 1 M CPB (pH 5). The dash line denotes the peak at 1654 cm⁻¹. **c** Comparison of N-R6-C, N-R12-C and N-R12-C(xA) in three sections along the channel. Peak features were specifically evaluated within the amide I region. **d** Raman results of N-R12-C fiber under various negative pressures. The solid line denotes the peak at 1664 cm⁻¹. **e** β-sheet contents of N-R12-C fiber assembled inside the microfluidic device under various negative pressures. Peak fitting was carried out to calculate the β-sheet contents after deconvolution within the amide I region. Data are presented as mean values ± SD; $n = 3$. **f** Comparison of Raman signals for N-R6-C, N-R12-C and N-R12-C(xA) fibers formed by biomimetic spinning and manual drawing. Source data are provided as a Source Data file.

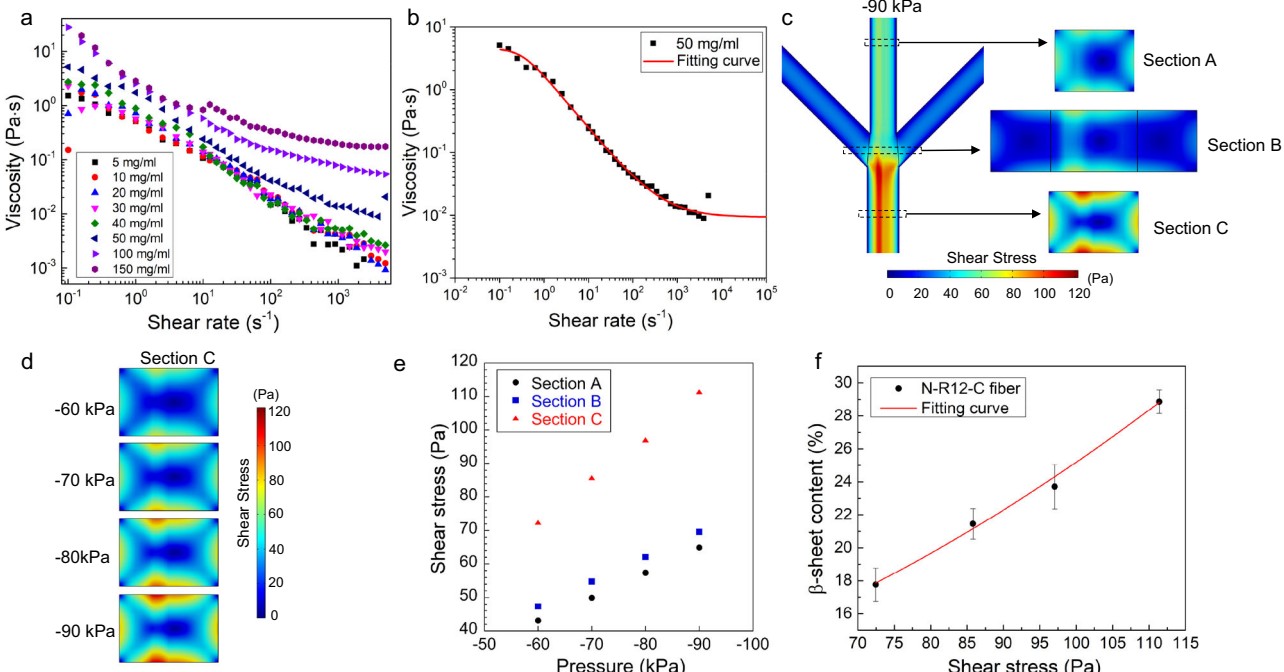

**Fig. 4 | Numerical simulation of shear stress across the length of microfluidic channel. a** The viscosity-shear rate relationship of N-R12-C solution at various concentrations from 5 to 150 mg/ml. **b** Shear-thinning behavior of N-R12-C solution at a concentration of 50 mg/ml with the result fitted by the Carreau model. **c** Simulated shear stress distributed along the channel to show the difference in each section. **d** Comparison of shear stress s in section C when changing the negative pressure from −60 to −90 kPa. **e** The value of shear stress estimated in sections A, B and C under different negative pressures. **f** The β-sheet contents of N-R12-C fiber under specific shear stress estimated from the corresponding negative pressure. The red line shows the fitting result by using the quadratic function. Data are presented as mean values ± SD; $n = 3$. Source data are provided as a Source Data file.

To test the threshold hypothesis for triggering β-sheet emergence, microfluidic assembly of N-R12-C was performed under varying magnitudes of negative pressure (Fig. 3d). At the lower levels (−40 to −50 kPa), fibers were not obtained, with the protein instead forming aggregates. In contrast, from −60 to −90 kPa hierarchical fibers were formed, with a positive correlation between the magnitude of negative pressure and the proportion of β-sheet conformations (Fig. 3e). Interestingly, −60 kPa was confirmed to be the threshold pressure for the formation of silk fiber, which obtained an averaged β-sheet content of 17.8%. With regard to the increase of pressure, the β-sheet content was gradually increased by 21.5% (at −70 kPa), 23.7% (at −80 kPa) and 28.9% (at −90 kPa). It was assumed that more β-sheets could be induced within silk fibers if the pressure reached beyond −90 kPa that was limited by our experimental system. This result indicates the potential for tuning fiber structure via subtle changes in the applied pressure.

Upon exiting a spider's spinneret, dragline silk is usually subjected to further drawing to achieve higher degree of crystallization and orientation. This natural behavior represents the same principle as applied in forcibly reeled silkworm silks[33] and postdrawing as-spun fibers[34]. We thus investigated the relative levels of β-sheet formation that could be achieved through our microfluidic device versus manually drawn fibers (Fig. 3f). Although precise pH gradients and shear cannot be induced using the manual drawing method, the elongation force generated from manual drawing is generally considered strong and responsible for sufficient induction of β-sheets[34]. While the microfluidic system could drive the hands-free, biomimetic assembly of recombinant silk fibers, the resultant fibers displayed a lower β-sheet content (Supplementary Table 2) when compared to fibers produced by manually drawing phase separated MaSp2 (which produced β-sheet even in the shorter N-R6-C constructs). This could reflect the greater degree of mechanical deformation afforded by externally stretching the fibers in air, along with dehydration effects. In

support of Raman results, wide-angle X-ray scattering (WAXS) results suggested the same tendency regarding the content of β-sheet nano-crystals in MaSp2 fibers formed by different spinning approaches (Supplementary Fig. 6). After normalization of peak (020)[35], the intensity at peak (010) suggested that the relative crystallinity of silk fibers prepared by manual drawing was positively related with the number of polyalanine blocks within the repetitive domain. In the absence of polyalanine blocks, the crystallinity of N-R12-C(xA) fiber made by sufficient manual drawing was still lower than that of N-R12-C fiber made by microfluidic spinning. In any case, these results suggest a further possibility of fine-tuning the properties of biomimetic silk by combining aqueous spinning methods with mechanical post-processing steps.

## Computational analysis

To elucidate the MaSp2 fiber assembly mechanism, we performed numerical simulations on the distribution of components and inter-play of physical forces during operation of the device. Before modelling, the rheological experiments were performed to probe the viscoelastic properties of N-R12-C solution under concentrations from 5 to 150 mg/ml (Fig. 4a, b). In line with the results reported for other recombinant (NT2RepCT) and native spidroins[36,37], N-R12-C solution exhibited a typical shear-thinning behavior with decreased viscosity in response to an increase in shear rate. Given a shear rate of $1\,s^{-1}$ at 25 °C, the shear viscosity of native spidroins was in the order of $10^3$ Pa s, much higher than that of N-R12-C (all below 10 Pa s) and NT2RepCT (below 1 Pa s). This significant difference can be interpreted by the much higher concentration and molecular weight of native spidroins in comparison to these of recombinant spidroins. To directly compare the shear viscosity between recombinant spidroins under the same concentration (such as 100 mg/ml), N-R12-C exhibited remarkably higher shear viscosity (~2 Pa s), more than 20 times higher than that reported for NT2RepCT (0.05–0.1 Pa s), which could be attributed to

the different number of repeat units (12 repeats vs 2 repeats). In this work, the alanine residues in the repetitive region were found to contribute significantly to the viscosity of spidroins as evidenced by the lower viscosity (apparent to the naked eyes) of N-R12-C(xA) when compared with N-R12-C.

The shear-thinning mechanism is not fully understood, and two theories were proposed based on the colloid and polymer systems[38,39]. In the colloid system, the phase separation during flow caused the shear thinning behavior, a phenomenon attributed to the disentanglement of molecular chains in the polymer system. As N-R12-C is a biopolymer and can be sheared to induce phase separation, thus its shear thinning behavior complies well with both theories. Using the Carreau model to fit the viscosity-shear rate curves under various concentrations, the non-Newtonian 3D model was well established. In this way, the distribution of various ions (citrate, phosphate, chloride and sodium ions), protein concentration, shear and elongational rate/stress along the channel were modeled (Supplementary Figs. 7–9).

In terms of physical forces generated within the device, as anticipated from the design, elongational flow effects within the device were minimized, with negligible values in the non-convergent channels of section A and C (Supplementary Fig. 9). Due to the intersectional geometry, there might be certain elongation stress applied to the material protein in the threshold region between sections A and B, but we considered its effect trivial after confirming that the local shear stress was more than nine times higher than the elongation stress.

On the other hand, shear effects were found to exert predominant effects during fiber assembly throughout the channel, evidenced by the colocalization of the shear-rich area (Fig. 4c) and MaSp2 (Fig. 2a, b) consistently from section A to C. Corresponding to specific pressure, the distribution and value of shear stress/rate within channel (Fig. 4d, e and Supplementary Fig. 10) was effectively predicted by our model. As indicated above, −60 kPa was the critical pressure required for silk fiber assembly inside the microfluidics, which is equivalent to the threshold shear stress and rate of 72 Pa and 52282 s$^{-1}$. It is worth noting that the estimated shear rate (52282 s$^{-1}$) required for MaSp2 assembly was one order of magnitude higher than that reported for native spidroins (1500–3500 s$^{-1}$)[40,41]. This result can be explained by the

difference between recombinant and natural spidroins in terms of concentration, viscosity, and molecular weight (length of repetitive domain) as well as the spinning and boundary conditions defined in computational modeling. The emergence of shear-induced β-sheets was observed for only three-domain MaSp2 fiber with at least 12 polyalanine blocks, and the β-sheet content within the N-R12-C fiber was estimated to range from 18% to 29% (Supplementary Fig. 11), as modulated by varying the shear stress from 72 to 111 Pa (Fig. 4f). Distinctly, by using molecular dynamics simulations of shear flow, MaSp1 fibers having at least 6 polyalanine blocks without the CTD and NTD were able to form β-sheet-rich structures in the presence of shear stress up to 300–700 MPa[42].

## Silk self-assembly mechanism

LLPS is the initial step during silk assembly and remarkably critical for the following nanofibrillization that is responsible for hierarchical organization (Supplementary Discussion). In agreement with our previous study[12], the full domain was necessarily required to enable both LLPS and nanofibrillation for MaSp2 fiber formation. To date, the underlying mechanism to drive LLPS remains largely unknown. As recently reported[43,44], Tyr and Arg residues functioned like molecular "stickers" to induce the phase separation of spidroins through LLPS. In this work, Ala residues, particularly in a tandem format (i.e., poly-alanine blocks) were confirmed an irrelevant factor to affect LLPS by observing the spontaneous response of N-R12-C(xA) toward the trigger of kosmotropic anions (Supplementary Discussion). Likewise, nanofibrillation was not seen to be inhibited by the absence of polyalanine blocks with macroscopically indistinguishable hierarchical structures visualized in the N-R12-C(xA) fiber (Fig. 5a; Supplementary Fig. 4).

Without using the well-established liquid crystalline[14] and micelle[45] theories, LLPS was taken instead as a paradigm to appropriately understand the silk self-assembly process in our biomimetic spinning system (Supplementary Discussion). Native-like triggers were sequentially introduced to the microfluidic channel for the generation of ion and pH gradient, allowing the induction of MaSp2 into solid nanofibrils-containing fibers (Fig. 5b). Both chemical and physical triggers were needed and worked synergistically for fiber formation,

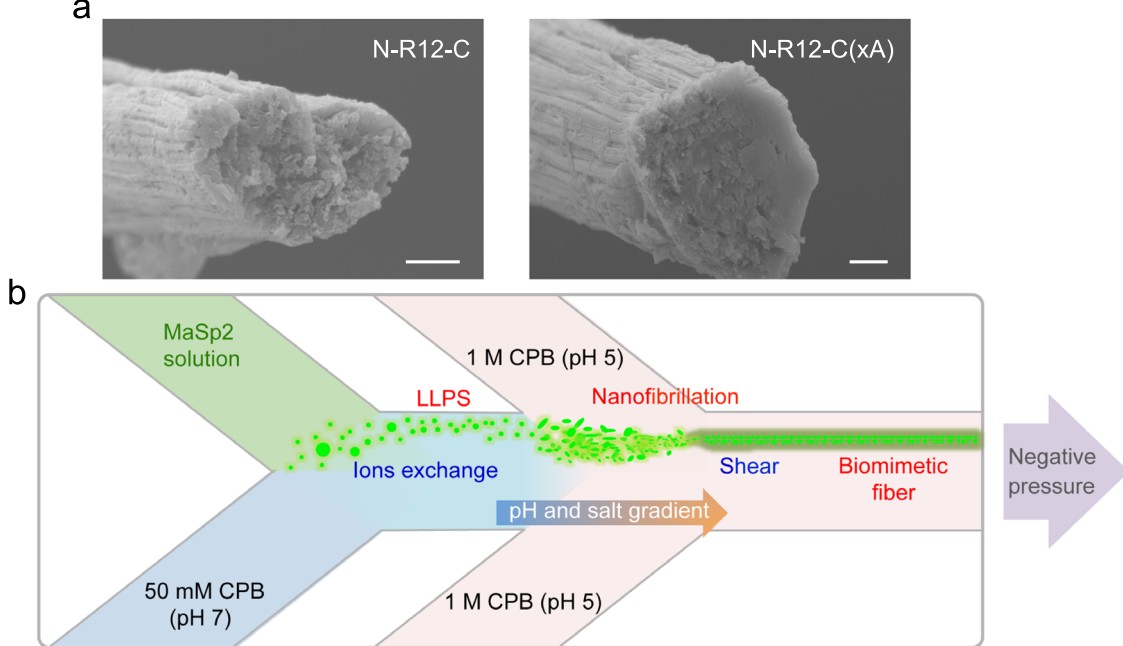

**Fig. 5 | Spider silk self-assembly mechanism from amino acids to functional triggers. a** Hierarchical structure of N-R12-C and N-R12-C(xA) fibers imaged by SEM. Four independent experiments were performed with similar results. Scale bar, 5 μm. **b** Schematic of spider silk self-assembly from LLPS to nanofibrillation to fiber formation triggered by native-like conditions.

however the latter played a dominant role for the transition of secondary structure. It is flexible to apply the microfluidic system to control the shear, which was highlighted in this work to investigate its impact on crystallization. By regulating the shear stress, it is promising to fabricate artificial silk fibers with subtly tuned β-sheet structure.

## Discussion

In summary, by using a microfluidic device, we report a biomimetic strategy in probing the β-sheet development under native-like assembly process from LLPS to nanofibrillization to hierarchically organized fiber. This work provides a basis for the rational design of biomimetic spinning with regard to domain structure, microfluidic fabrication, shear threshold and hierarchical reconstruction. Future efforts will focus on investigating the silk assembly of combined MaSp1 and MaSp2 spidroins under biomimetic spinning conditions. The heterodimerization formed by CTD and/or NTD will be comprehensively evaluated to better understand the interaction between spidroins in natural spinning process. Moreover, specific amino acid residues in the gene sequence will be edited/substituted to explore the underlying mechanism for LLPS.

## Methods

### Protein preparation

Expression of recombinant MaSp2 was performed as previously described[12]. Briefly, the pET15-derived constructs were transformed into *E. coli* BL21(DE3) cells, grown overnight as 100-mL preculture in LB medium with ampicillin (Amp; 100 μg/mL) at 37 °C with shaking at 180 rpm, then inoculated into 2 L LB-Amp, grown at 37 °C until OD600 ~ 1.0, after which the temperature was lowered to 20 °C and protein expression induced overnight with 0.4 mM isopropyl-beta-d-thiogalactopyranoside (IPTG). For protein purification, cell lysis was carried out by freeze-thawing the bacterial cell pellet that had been completely resuspended in 20 mM Tris-HCl pH 7.5, 20 mM imidazole, 0.5 M NaCl supplemented with 1% Triton X-100 (Sigma-Aldrich), 0.5 mg/ml chicken egg lysozyme (Wako), 250 units TurboNuclease (Accelagen), and cOmplete EDTA-free protease inhibitor tablet (Roche). The lysate was spun down and the supernatant fraction applied to a His-TRAP FF column (2x5ml in tandem) (Cytiva), washed with 20 mM Tris-HCl pH 7.5, 20 mM imidazole, 0.5 M NaCl, and eluted with 20 mM Tris-HCl pH 7.5, 250 mM imidazole, 0.5 M NaCl. The eluate was subjected to buffer exchange against 20 mM Tris-HCl pH 7.5, 0.15 M NaCl using a VivaSpin spin concentrator with 10 kDa MWCO (Sartorius). The poly-histidine tag was removed by overnight thrombin digestion (Sigma-Aldrich) at 4 °C, followed by concentration using VivaSpin. For x-R6-C, N-R6-C, N-R12-C and N-R12-C(xA), a final round of purification was carried out through LLPS, whereby 1.0 M KPi, pH 7.0, was added until two stable liquid phases were formed, followed by centrifugation and retrieval of the high-density phase, which contained highly purified MaSp2, which was then diluted in 20 mM Tris-HCl, 0.15 M NaCl. The protein concentration was measured by absorbance at 280 nm using a NanoDrop instrument (Thermo Fisher Scientific). The protein concentration of 50 mg/ml was applied for the following experiments unless otherwise indicated.

### Protein electrophoresis

SDS–PAGE was performed on 4–20% mini-PROTEAN TGX precast gels (Bio-Rad) under denaturing conditions according to the manufacturer's recommendations. In the case of nonreduced samples, β-mercaptoethanol (βME) was omitted from the sample buffer, and the samples were not heated prior to electrophoresis. Gels were stained with 0.1% Coomassie R-250 in 40% ethanol and 10% acetic acid.

### Fabrication of the microfluidic device

The microfluidic device was prepared basically according to the widely reported[20] conventional process, from mask design, mold fabrication,

and PDMS casting to chip integration. After degassing overnight, SU-8 (3050) was poured onto a clean wafer for spin coating for 30 s at 1650 rpm. The wafer with uniformly dispersed SU-8 was transferred to the oven to prebake for 20 min at 95 °C and then slowly cooled for 10 min. The photomask with specific channel patterns was closely aligned to the SU-8 wafer for photolithography, followed by postbaking and developing to retain only the pattern on the silicon wafer. The mold was successfully fabricated after stepwise high-temperature baking from 150 to 200 °C and silanization overnight by using trichloro(1H, 1H, 2H, 2H-perfluorooctyl)silane (Sigma Aldrich). Subsequently, a PDMS slab containing channels was obtained by casting a silicone elastomer (SYLGARD 184, Dow Corning) with its curing agent on the mold. Three inlets and one outlet with hole diameters of 1.5 mm and 4 mm, respectively, were punched. Without permanent binding treatment, the PDMS slab was attached to the coverslip to obtain the final microfluidic device. In this study, the microfluidic device was operated under native pressure to enable a strong and stable interface between PDMS and the coverslip to avoid leaking risk. The coverslip was selected rather than a relatively thick glass slide mainly because of the conditions required by Raman spectroscopy and CLSM.

### Silk assembly within the microfluidic device

The PDMS was tightly attached to the coverslip without oxygen plasma treatment for permanent binding. No leaking of fluids was observed during or after the experiment. A pressure control system consisting of an electropneumatic regulator (SMC ITV0090-2CL, USA) and microcontroller (Arduino, Italy) was established to drive the fluids inside the microfluidic channels. The real-time pressure (up to − 90 kPa) applied to the outlet was accurately monitored by the integrated Spyder software. Various attempts were made to achieve reliable and reproducible processing for silk assembly within the microfluidic device, as shown in Supplementary Movie 1. The sequence of steps was as follows: (1) turn on the vacuum pump (with adjustable pressure output) and attach the connected tubing to the device outlet; (2) pipet 15 μl of 1 M CPB (pH 5) into inlet 3; (3) pipet 15 μl of 50 mM CPB (pH 7) into inlet 2; (4) pipet 1.5 μl of 50 mg/ml MaSp2 into inlet 1; and (5) disconnect the vacuum pump tubing. The protocol was performed rapidly and was essentially complete upon pipetting of the protein solution. Ultimately, recombinant spider silk was assembled inside the microfluidic channel. The device was used to check the in situ MaSp2 morphology with various microscopes and characterize the secondary structure by Raman spectrometry.

### Optical microscope

The sample morphology was directly observed through the outlet of the microfluidic device by using an Olympus BX53 microscope. To visualize the sample within the channel, the microfluidic device was reversely placed on the glass slide before transferring to the optical microscope.

### Confocal laser scanning microscopy

MaSp2 N-R12-C was first labeled with DyLight-488 and then subjected to microfluidic spinning. Afterward, the microfluidic device was immediately transferred to an inverted Zeiss LSM 700 confocal scanning microscope. With an excitation length of 448 nm, 40x numerical aperture and 60x water immersion objectives were utilized to image MaSp2 inside the channel. Supporting movies were made by the stitch of each single captured image to clearly demonstrate the whole spinning process from the inlet to outlet. Image stitching was performed using ImageJ with the MosaicJ plugin[46]. To observe the nanofibrils, additional experiments were carried out to recover the freshly formed fibers from the microfluidic device and place them in a hypotonic environment (Milli-Q water), which caused the fibers to swell. There were localized areas where the continuity of the fiber was broken to allow the visualization of high-aspect ratio nanofibrils.

High-resolution 3D imaging of biomimetic fibers within the microfluidic channels (using MaSp2 N-R12-C labeled with DyLight-488) was carried out using a TCS SP8 confocal microscope platform (Leica) under the Hyvolution2 regime. Imaging was performed using a 20x/0.75 HC PL APO CS2 objective, HyD 2 hybrid detector (Ex = 488 nm; Ex = 508–532 nm), and z-stacking with a 0.27 μm step size. Image deconvolution was achieved using Huygens Professional software (Scientific Volume Imaging).

## Scanning electron microscopy

The surface structures of MaSp2 droplets and fibers were detected by using a JEOL-6000 instrument at an accelerating voltage of 15 kV. To detect samples within the microfluidic device, PDMS was gently detached from the device to allow part of the samples to be retained on the coverslip. After thorough drying, the coverslip was directly fixed to an aluminum stub with conductive carbon tape and then subjected to gold sputtering for 2 min. It is noteworthy that the sample on the coverslip was covered with a thin layer of salts from the buffer solution. Samples could be damaged or partly dissolved after rinsing with Milli-Q water. Thus, no extra treatment was performed to remove the salt, and the overall morphology of samples detected by SEM was quite similar to that detected by CLSM. To observe the cross-section, MaSp2 fibers prepared by manual drawing were immersed in liquid nitrogen for 1 min and then cut by a blade.

## Raman spectra

Raman measurements were carried out with a JASCO NRS-4100 confocal microscope with a 532 nm laser. Using a 100x oil objective, the laser beam was focused on the center of the MaSp2 samples through the coverslip side, as shown in Fig. 3a. Spectra were recorded in a wavenumber range from 500 to 2000 cm$^{-1}$. To detect the minor shift in wavenumber, the grating was set as 1800 grooves mm$^{-1}$ with a slit size of $10 \times 8000$ μm, providing a resolution as high as 0.8 cm$^{-1}$. The beam intensity of 16.9 mW and exposure time of 300 s were optimized to obtain a sufficient signal-to-noise ratio. Under the protection of the buffer medium surrounding the MaSp2 samples, no local heating effect was induced by the set laser power. Consistent results were obtained by repeating the measurement at least five times under the same conditions to ensure data reliability. As a control, MaSp2 was gently mixed with 0.5 M CPB (pH 7 or 5) in a customized chamber made from vinyl adhesive tape on a glass slide. To create a similar background as inside the microfluidic channel, native dragline silk from *Trichonephila clavata* and manually stretched MaSp2 fibers were immersed in 1 M CPB (pH 5) for Raman experiments.

## Wide-angle X-ray scattering

The crystalline structure of MaSp2 fibers was determined by WAXS at SPring-8 synchrotron in Harima, Japan. The BL05XU beamline was selected to have X-ray energy as high as 15 KeV at a wavelength of 0.08 nm. The sample-to-detector distance was measured as 262 mm with an exposure time of 1 s set for each sample. 1D profiles were converted using Fit2D software from 2D patterns. Analogous to the SEM operation, PDMS was carefully peeled from the microfluidic device to better subtract the glass background in the following data analysis. After either biomimetic spinning or manual drawing, MaSp2 fibers were rinsed thoroughly with Milli-Q water to remove salts remaining on the fiber surface to avoid the adverse X-ray signal generated from salt crystals.

## Rheological analysis

The rheological behavior of MaSp2 (N-R12-C) was measured by an Anton Paar MCR 502 rheometer in rate-controlled mode. The experiment was conducted using a cone-and-plate fixture, where the cone diameter and angle were 8 mm and 2°, respectively. To prevent water evaporation from the sample, the water-wetted tissues were placed around the plate with a cover to maintain the saturated humidity (Supplementary Fig. 12). 7.1 μl of MaSp2 was pipetted onto the plate at various concentrations from 5 to 150 mg/ml. When it comes to the set temperature (25 °C), the sample was subjected to the shear rate ramp from 0.1 to 5000 s$^{-1}$ to measure the viscoelastic property of MaSp2 dope. The interval time was set as 120 s under shear rate from 0.1 to 10 s$^{-1}$, 60 s from 11 to 100 s$^{-1}$ and 25 s from 101 to 5000 s$^{-1}$. It is worth noting that the ultrahigh shear rate region (above $10^4$ s$^{-1}$) is a region that cannot be measured by a rotational rheometer.

## Numerical model

Based on the rheological behavior of MaSp2 (N-R12-C) solution, the related viscosity-shear rate curve was fitted using Carreau model[47] for non-Newtonian simulation. To account for the effect of MaSp2 concentration on viscosity, we fit the Carreau model as a function of concentrations to the viscosity as below:

$$\mu_{eff}(\dot\gamma) = \mu_{\inf} - (\mu_0 - \mu_{\inf})(1 + (\lambda\dot\gamma))^{\frac{n-1}{2}},$$

where $\mu_{\inf}$, $\mu_0$, $\lambda$, and $n$ are respectively defined as

$$\mu_{\inf}(c) = \mu_{CPB}(a_{\inf}c + 1),$$

$$\mu_0(c) = \mu_{CPB}(a_o c^{k_0} + 1),$$

$$\lambda(c) = \exp(a_\lambda c) - 1,$$

$$n(c) = a_n c^{k_n}.$$

Here $c$ represents the dimensionless concentration of MaSp2 normalized with 50 mg/ml. Other parameters are summarized in Supplementary Table 3.

The adventive diffusion equation was used for the molecular diffusion of MaSp2 and ions at infinite dilution. The channel geometry is shown in Fig. 1c. We computed the numerical model using COMSOL Multiphysics® 6.1 with interfaces of laminar flow and transport of diluted species. The software generated rectangular elements on the top surface of the channel and swept them to the bottom surface, yielding 134,151 quadratic hexahedral elements. The physical parameters of the numerical model are summarized in Supplementary Table 3. To simplify the numerical model, constant diffusion coefficients in the entire calculation were utilized. Therefore, the diffusion constant of MaSp2 in 1 M CPB (pH 5) was determined by dynamic light scattering (DLS) using a Zetasizer Nano-ZS (Malvern Instruments). The viscosity of 1 M CPB (pH 5) was measured by a vibration-type viscometer (VM-1G, CBC Co., Ltd.).

## MaSp2 fiber prepared by manual drawing

Artificial spider silk fibers were prepared by manually drawing MaSp2 out of 1 M CPB at pH 5. A 0.5 μl MaSp2 solution was first pipetted onto the cell culture dish at a concentration of 100–130 mg/ml, and then 4 μl CPB was immediately added over the MaSp2 solution. A sharp tweezer was taken to pull out one end of condensed turbid proteins to some extent, and subsequently, another tweezer was taken to uniaxially extend the fiber at the other end to the maximum length in the air.

## Reporting summary

Further information on research design is available in the Nature Portfolio Reporting Summary linked to this article.

## Data availability

The data supporting the findings of this study are included in the published article and its supplementary information files. The simulation data generated in this study have been deposited in the GitHub repository (https://github.com/Dubbing173/Simulation-data.git). Source data are provided with this paper.

## Code availability

The code used for simulation is available at https://doi.org/10.5281/zenodo.10408439[48].

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

## Acknowledgements

This work was financially supported by the Grant-in-Aid for Early-Career Scientists from JSPS (Grant No. 21K15063 to J.C.), National Natural Science Foundation of China (Grant No. 52303281 to J.C.), Departmental General Research Fund (Grant No. P0046210 to J.C.), the Japan Science and Technology Agency Exploratory Research for Advanced Technology (JST-ERATO; Grant No. JPMJER1602 to K.N.), CREST (to K.U. and K.N.), COI-NEXT (to K.N.), and MEXT Data Creation and Utilization-type MaTerial R&D project (to K.N.).

## Author contributions

J.C. and K.N. designed the research. A.D.M. and J.C. prepared and characterized the recombinant spidroins. J.C., A.T., and H.S. designed and fabricated the microfluidic chip. J.C. and A.D.M. performed the CLSM measurement. H.M. conducted the WAXD measurement. K.T., Y.T., M.K., and K.U. analyzed the rheological data. A.T. and H.S. carried out the simulation. J.C. performed the other experiments. J.C., A.D.M., and K.N. prepared the first draft of the manuscript and all authors contributed to the final draft.

## Competing interests

The authors declare no competing interests.
