## [Peer Review File · Nature Communications]

Replicating shear-mediated self-assembly of spider silk through microfluidicsREVIEWER COMMENTS

Reviewer #1 (Remarks to the Author):

Chen et al. present a study on the assembly of spidroin to produce spider silk. While the subject matter is intriguing, the overall quality of the paper does not meet the standards expected by Nature Communication. Some of the results lack substantial evidence to support the claims made in the main text. The primary objective of the paper is to offer deeper insights into shear-induced crystallization, as indicated by the title. However, the authors neglect to perform a fundamental characterization of the shear field and fail to analyze the fluids before and after assembly for shear properties. Instead, the only insight provided is through a single-phase Newtonian simulation, which raises doubts about its relevance considering the presence of two phases and the oversimplification of the fluid as Newtonian. Moreover, based on the methods described (e.g., Video 1), the experimental protocol appears somewhat unrefined, raising specific concerns about the inhomogeneity observed in the flows of different streams.

Some specific comments are listed below:

1. Line 146: In the laminar flow regime, the two streams coexist and occupy half of the channel width, specifically in Section A of the device. This arrangement minimizes the surface area between the two phases, namely the precursor spidroin in (1) and kosmotropic ions in (2), resulting in minimal surface energy and no energy gain for the system to undergo liquid-liquid phase separation (LLPS). It is rather surprising to observe that droplets spontaneously form under these conditions. Could the authors please provide further details regarding the process of LLPS occurring in Section A of the microfluidic system?
2. Line 156: Extensional flow can also be induced in convergent geometries.
3. Line 196: Concerning the term "nano fibril network," there is insufficient evidence to classify it as a fibril network. Figure 1(c) clearly shows the presence of a network structure, but the nature of the fibril-like components is uncertain.
4. Line 335: The authors assume a one-phase and Newtonian flow for the simulation. How accurate is this Newtonian assumption? The authors present a rheological graph in the Supplementary material without providing any information on how the results were obtained, lacking necessary specifications. It is recommended that the authors perform steady-state shear rheology to obtain valuable information about the fluid properties.
5. Figure 4(a) is misleading. For a Newtonian fluid, the shear rate should increase linearly from the centerline to the walls. However, the figure shows the opposite trend. Is this discrepancy due to the rectangular cross-section of the channel? Additionally, it appears that the stress is generally higher in the central-top part of the device in Figure 4(a) compared to the two sides. Can the authors explain this observation considering the identical dimensions of the channels? Does it imply different average velocities for the flow of the fibrillation trigger solution and the LLPS part?

6. Video 1 appears to be faster than real-time. It is recommended that the authors provide information specifying the speed of the video for better understanding.

7. Is there a specific reason for using -90 kPa pressure in this work?

Reviewer #2 (Remarks to the Author):

The manuscript described investigations of the assembly mechanism of recombinant spidroin constructs, using a microfluidic device that aims to mimic natural triggers of spider silk assembly. The manuscript is clearly written and the results shown give more insights into the silk assembly mechanism.

Specific comments, minor and major, in order of appearance of the text:

1. Introduction, second paragraph: what do you mean by “by and large”?
2. Introduction, third paragraph: What do you mean with “have as a rule not been carried out”?
3. Please avoid the use of the term “full-domain” (used at several places). It gives the impression of being the native full-length spidroin.
4. Results, line 206: Explain how “effective fiber formation” was analyzed, evaluated and judged.
5. Results, line 238-240. Specify where in the device the samples were taken.
6. Results, line 248: What do you mean with “wild-type”? Native spidroin?
7. Results, Raman: Include references for the assignment of conformations in the amide I region.
8. Figure 3. Include helplines for both alpha and beta structures. Preferably also helplines for facilitating comparison of intensities.
9. Results, line 290-294. Include references for the assignment of conformations in the amide I region.
10. Results, WAXD: Explain how the results were evaluated and include a reference for the theory behind it.
11. Methods: Include a reference for how expression and purification were performed, and give a brief overview also here (main methods and conditions).
12. Methods: What do you mean by “purification by LLPS”? Precipitation with phosphate?
13. Methods: Which protein concentrations were used?

Reviewer #3 (Remarks to the Author):

This manuscript described the fabrication of artificial spider silk based on recombinant MaSp2 spidroin through a microfluidic device. The author focused on the effect of shear stress, pH and polyalanine blocks on the β -sheet content and morphology of the obtained fiber. However, there is a lack of the most important mechanical property of this fiber, and some of data and discussions were inadequately

explained and insufficient to support conclusions. Therefore, this manuscript in the present form is unsuitable for publication in Nature Communications. It may be resubmitted to the journal after rewriting.

1. The author claimed that replicating the mechanical properties of the natural spider silk is still a challenge, how about the mechanical properties of the fiber fabricated in this manuscript. The systemic test, discussion, and comparison of the mechanical properties should be provided.
2. Line 80-97: Many factors will affect the structure, morphology, and mechanical property of artificial spider silk, how to give an overall consideration?
3. Line 113: As the author claimed that reference 22 had shown fiber formation using full-domain ADF3 spidroin through sequential introduction of phosphate ions and pH decrease in a microfluidic device, what is the difference and advantage of this manuscript compared to reference 22?
4. Line 117-119: The description “quantitative analyses of the forces generated within the microfluidic channels during sample flow, and their influence on the conformation of the silk proteins, have as a rule not been carried out in rigorous detail” is incorrect, the relevant studies have been reported, for example: Adv. Mater. Technol. 2021, 2100124; ACS Sustain. Chem. Eng., 2019, 7, 14765-14774. Hence, the author should be discussed carefully the advantage of this manuscript.
5. Line 154: Why did the author design section C with the same width (80 μm)?
6. Line 232, Fig. 2: The hierarchical structure is not clear as shown in Fig. 2d. In addition, there is a big saltation of the shear stress at section B, how to affect the morphology, structure and property of the fiber.
7. Line 308-310: The quantitative analysis of β -sheet content should be provided.
8. Line 317: How to regulate the pH gradient of the manual drawing method. The detailed information should be added.
9. Line 334-335: During the simulation, only Newtonian laminar flow is considered. However, there are laminar and turbulent flows in the microfluidic device. Hence, the reliability of this simulation result is not high.
10. Line 393, Fig. 5: What is the meaning of the hierarchical structure, and how to distinguish this structure? What does the scale bar represent?

Reviewer #4 (Remarks to the Author):

RESPONSES TO REVIEWER COMMENTS

Reviewer #1

Comment 1: Line 146: In the laminar flow regime, the two streams coexist and occupy half of the channel width, specifically in Section A of the device. This arrangement minimizes the surface area between the two phases, namely the precursor spidroin in (1) and kosmotropic ions in (2), resulting in minimal surface energy and no energy gain for the system to undergo liquid-liquid phase separation (LLPS). It is rather surprising to observe that droplets spontaneously form under these conditions. Could the authors please provide further details regarding the process of LLPS occurring in Section A of the microfluidic system?

Response: Thank you for your question. Outside the microfluidic chip, spontaneous formation of droplets can be observed by the confocal laser scanning microscope (CLSM) when mixing the protein solution with 0.5-1 M CPB at pH 7. However, inside the microfluidic chip, the whole process for the fiber formation is too fast to allow *in situ* observation under the experimental condition (see Movie 1). The interactions in the interface of laminar flow and some turbulence may lead to the induction of LLPS. More details have been added to the revised manuscript to elaborate the process of LLPS in Section A.

In Page 8, Line 230-235: “This result was in agreement with the dynamic formation and fusion behavior of droplets as shown in Fig. 1c. The fact that MaSp2 was able to undergo extensive LLPS despite the relatively limited surface contact between the buffer phase and the kosmotropic trigger due to laminar flow indicates that the initial conditions were close to the phase separation boundary, which ensured prompt phase separation when the two streams came into contact.”

Comment 2: Line 156: Extensional flow can also be induced in convergent geometries.

Response: Yes, the extensional flow can be induced in the convergent geometries. Our simulation results showed the elongational force was mainly generated in section B. Inside the natural spider silk gland, the duct is progressively narrowed down toward the spinneret, enabling a strong extensional flow to align the fiber for better mechanical property and higher crystallization. However, this work is mainly focused on the exploration of shear effect on the silk fiber formation and crystallization, therefore a widening width from section B to section

C was intentionally designed to minimize the effect of elongational force. We have added more details to the revised manuscript as follows.

In Page 5, Line 161-167: “It is important to state that we chose a simplified design where each individual channel has a uniform cross-section, and an increase in width from section B (60 μm) to C (80 μm), in order to focus on the effects of shear in modulating fiber assembly and crystallization with minimized effects of extensional flow. This is contrast to the more complex (i.e., convergent) geometries found in the natural spinning ducts, where the duct is progressively narrowed down toward the spinneret to enable a strong extensional flow for the alignment of silk molecules for better orientation.”

Comment 3: Line 196: Concerning the term "nano fibril network," there is insufficient evidence to classify it as a fibril network. Figure 1(c) clearly shows the presence of a network structure, but the nature of the fibril-like components is uncertain.

Response: To avoid misunderstanding, the term “nanofibril network” was revised as “network”. In fact, the “network” and “nanofibril” are closely associated with each other and can be respectively considered as 2D and 1D structure under different processing conditions. More descriptions have been added to the revised manuscript as follows.

In Page 8, Line 235-242: “In Fig. 1c, the network structure was formed under the condition that one drop of N-R12-C solution was placed on the cover slip and then another drop of buffer (i.e., 0.5 M CPB at pH 5) was added onto the N-R12-C solution. Therefore, a network structure was formed along with the random spreading of buffer. However, inside the microfluidic channel, the flows were driven unidirectionally and the nanofibrils were formed aligning to the length of fiber with orientation. In some sense, 2D network structure can be transformed into 1D nanofibrils after uniaxial stretching of the network in a proper manner.”

Comment 4: Line 335: The authors assume a one-phase and Newtonian flow for the simulation. How accurate is this Newtonian assumption? The authors present a rheological graph in the Supplementary material without providing any information on how the results were obtained, lacking necessary specifications. It is recommended that the authors perform steady-state shear rheology to obtain valuable information about the fluid properties.

Response: Thank you for the question and suggestion. We have conducted the steady-state shear rheology by using Anton Paar MCR 502 Rheometer (New Supplementary Fig. 12) to

determine the viscoelastic property of MaSp2. Various concentrations of MaSp2 (N-R12-C) were tested to show the viscosity-shear rate relationship and then the corresponding curves were fitted using Carreau model for the simulation. New Fig. 4a and 4b were made and added to the revised manuscript. In line with another recombinant (T. Arndt, *ACS Biomater. Sci. Eng.* 2021, 7, 462-471) and native spidroins (N. Kojic, *J. Exp. Biol.* 2006, 21, 4355-4362), the shear-thinning behavior of N-R12-C reflected a non-Newtonian flow. Then we correct the simulation with a non-Newtonian model and based on simulation results new Fig. 4 were made in placement of the old one. The discussion on the rheological property and non-Newtonian simulation of MaSp2 have been added in the revised manuscript.

In Page 14, Line 405-432: “Before modelling, the rheological experiments were performed to probe the viscoelastic properties of N-R12-C solution under concentrations from 5 to 150 mg/ml (Fig. 4a and b). In line with the results reported for other recombinant (NT2RepCT) and native spidroins,^{36,37} N-R12-C solution exhibited a typical shear-thinning behavior to see the decreased viscosity with regard to the increase of shear rate. Given a shear rate of 1 s^{-1} at $25 \text{ }^{\circ}\text{C}$, the shear viscosity of native spidroins was in the order of $10^3 \text{ Pa}\cdot\text{s}$, much higher than that of N-R12-C (all below $10 \text{ Pa}\cdot\text{s}$) and NT2RepCT (below $1 \text{ Pa}\cdot\text{s}$). This significant difference can be interpreted by the much higher concentration and molecular weight of native spidroins in comparison to these of recombinant spidroins. To directly compare the shear viscosity between recombinant spidroins under the same concentration (such as 100 mg/ml), the N-R12-C was remarkably higher in shear viscosity ($\sim 2 \text{ Pa}\cdot\text{s}$), more than 20 times higher than that of NT2RepCT ($0.05\text{-}0.1 \text{ Pa}\cdot\text{s}$) due to different numbers of repeat unit (12 repeats vs 2 repeats). In this work, the alanine residues in the repetitive region were found to contribute significantly to the viscosity of spidroins as evidenced by the lower viscosity (apparent to the naked eyes) of N-R12-C(xA) when compared with N-R12-C.

The shear-thinning mechanism is not fully understood, and two theories were proposed based on the colloid and polymer systems^{38,39}. In the colloid system, the phase separation during flow caused the shear thinning behavior, which was considered as phenomenon due to the disentanglement of molecular chains in the polymer system. As N-R12-C is a biopolymer and can be sheared to induce phase separation, thus its shear thinning behavior complies well with both theories. Using the Carreau model to fit the viscosity-shear rate curves under various concentrations, the non-Newtonian 3D model was well established. In this way, the distribution

of various ions (citrate, phosphate, chloride and sodium ions), protein concentration, shear and elongational rate/stress along the channel were modeled (Supplementary Figs. 7-9).”

Anton Paar MCR 502 Rheometer

Supplementary Fig. 12: Experimental setup for the MaSp2 (N-R12-C) solution by using Anton Paar MCR 502 Rheometer. The tissue was wetted by the water and placed around the plate without contacting the sample. Finally, a cover was used to maintain saturated humidity of the sample.

Fig. 4: Numerical simulation of shear stress across the length of microfluidic channel. a, The viscosity-shear rate relationship of N-R12-C solution at various concentrations from 5 to 150 mg/ml. **b,** Shear-thinning behavior of N-R12-C solution at a concentration of 50 mg/ml with the result fitted by the Carreau model. **c,** Simulated shear stress distributed along the channel to show the difference in each section with the top (left) and cross-section (right) view. **d,** Comparison of shear stress s in section C when changing the negative pressure from -60 to

-90 kPa. **e**, The value of shear stress estimated in sections A, B and C under different negative pressures. **f**, The β -sheet contents of N-R12-C fiber under specific shear stress estimated from the corresponding negative pressure. The red line shows the fitting result by using the quadratic function.

Comment 5: Figure 4(a) is misleading. For a Newtonian fluid, the shear rate should increase linearly from the centerline to the walls. However, the figure shows the opposite trend. Is this discrepancy due to the rectangular cross-section of the channel? Additionally, it appears that the stress is generally higher in the central-top part of the device in Figure 4(a) compared to the two sides. Can the authors explain this observation considering the identical dimensions of the channels? Does it imply different average velocities for the flow of the fibrillation trigger solution and the LLPS part?

Response: Thank you for your comment. The left image in Fig. 4a is the top view of microfluidic channels. To avoid misleading, more information has been added to the caption. In section C, the channel width was increased from 60 μm to 80 μm , so the shear stress is higher in the central-top part of the device with a channel height of 62 μm . The simulation has been corrected as the non-Newtonian flow and the new results were shown in the new Fig. 4.

In Page 16, Line 467-469: “**c**, Simulated shear stress distributed along the channel to show the difference in each section with the top (left) and cross-section (right) view.”

Comment 6: Video 1 appears to be faster than real-time. It is recommended that the authors provide information specifying the speed of the video for better understanding.

Response: Thank you for your suggestion. We have shot a new video to show the real-time experimental process for better understanding and reproducing. The new video has been submitted as Supplemental Movie 1.

Cover page of the Supplemental Movie 1

Comment 7: Is there a specific reason for using -90 kPa pressure in this work?

Response: Thank you for this question. The main reason is that the electronic vacuum regulator (ITV0090-2CL) used in our pressure control system is limited by the working range from -1 to -100 kPa. For safety reasons, -90 kPa was applied as the highest pressure in this work. When the pressure is lower than -60 kPa, it is not likely to assemble the silk fiber in the microfluidic chip. Under the highest pressure of -90 kPa, the highest β -sheet content can be induced accompanied by the formation of silk fiber.

Reviewer #2

Comment 1: Introduction, second paragraph: what do you mean by “by and large”?

Response: Thank you for the question. The phrase “by and large” means “generally”. For better understanding, we have changed it as “generally” in the revised manuscript.

Comment 2: Introduction, third paragraph: What do you mean with “have as a rule not been carried out”?

Response: The phrase “as a rule” means “in general”. For better understanding, we have deleted it and changed the original sentence as “have not been well carried out” in the revised manuscript.

Comment 3: Please avoid the use of the term “full-domain” (used at several places). It gives the impression of being the native full-length spidroin.

Response: As suggested, we changed “full-domain” as “three-domain” throughout the main text in the revised manuscript.

Comment 4: Results, line 206: Explain how “effective fiber formation” was analyzed, evaluated and judged.

Response: Thank you for this good question. In our experimental condition, for the truncated constructs N-R6-x, x-R6-C and x-R6-x, they can only form large aggregates inside the microfluidic channel under all conditions. However, for the constructs with complete three domains (i.e., N-R6-C, N-R12-C and N-R12-C (xA)), they can all form a single continuous fiber above the pressure of -60 kPa. Otherwise, breaking fibers, fibrous aggregates or their mixtures will be formed below this threshold pressure. To make the phrase “effective fiber formation” clearer, more explanations were added to the revised manuscript as follows.

In Page 7, Line 214-217: “Conditions for effective fiber formation, which we define as the generation of a single continuous fiber inside the microfluidic channel, excluding fiber breakage, or formation of aggregates or their mixtures, were established empirically.”

Comment 5: Results, line 238-240. Specify where in the device the samples were taken.

Response: Thank you for the reminder. We have added the information in the revised manuscript as follows.

In Page 10, Line 270-273: “**d**, N-R12-C fiber assembled inside the channel in section C3 with a hierarchical organization. Scale bar, 10 μm . **e**, 3D reconstruction of N-R12-C fiber aligned along the channel in section C3. Scale bar, 5 μm .”

Comment 6: Results, line 248: What do you mean with “wild-type”? Native spidroin?

Response: To avoid misunderstanding, we have deleted the phrase “wild-type”. It is not native spidroin but the constructs of N-R6-C and N-R12-C. In the revised manuscript, the “wild-type” was changed as “constructs of N-R6-C and N-R12-C”.

Comment 7: Results, Raman: Include references for the assignment of conformations in the amide I region.

Response: As suggested, two references have been added to the revised manuscript with detailed discussion on the assignment of conformations.

In Page 10, Line 296-307: “Spectra collected from native spider dragline silk fiber (Supplementary Fig. 4) produced well-defined peaks at the expected locations of 1670 (amide I, C=O stretching), 1615 (Tyr), 1452 (CH_3 asymmetric bend, CH_2 bending), 1242 (amide III, N-H bend + C-N stretching), 1175 (Tyr), and $\sim 1078/1094\text{ cm}^{-1}$ (skeletal $\text{C}_\alpha\text{-C}_\beta$ stretching). Of these peaks, the amide I, amide III and skeletal C-C peaks are sensitive to changes in secondary structure, particularly with regard to β -sheet conformations³¹. Normally, the peaks within the amide I or III regions can be deconvoluted to calculate the secondary structure composition³². However, the PDMS and coverslip showed signals at 1266 and 1098 cm^{-1} , respectively, that noticeably overlapped with those from the dragline silk. Therefore, in the succeeding sections, we focused on the area surrounding the amide I region for the quantitative analysis of secondary structure content.”

Comment 8: Figure 3. Include helplines for both alpha and beta structures. Preferably also helplines for facilitating comparison of intensities.

Response: Yes. We have already used helplines to differentiate the alpha and beta structures. But the explanation of the helplines is missing in the figure captions. Detailed information has been added to the revised manuscript as follows.

In Page 12 Line 322-326: “The dash line denotes the peak at 1654 cm^{-1} . **c**, Comparison of N-R6-C, N-R12-C and N-R12-C(xA) in three sections along the channel. Peak features were specifically evaluated within the amide I region. **d**, Raman results of N-R12-C fiber under various negative pressures. The solid line denotes the peak at 1664 cm^{-1} .”

To show the difference or change in peak intensity, Fig. 3 was revised for better demonstration.

Fig. 3: Characterization of the β -sheet formation of MaSp2 fiber formed in the microfluidic device or by manual drawing. a, Secondary structure of the MaSp2 fiber determined by Raman spectroscopy immediately after fiber assembly inside the channel. **b**, Negative control experiment of N-R12-C and N-R12-C(xA) outside the microfluidic device. N-R12-C and N-R12-C(xA) were gently mixed with 1 M CPB at pH 5 and 7 on a cover slide at a 1:1 volume ratio. A positive control was conducted by immersing N-R12-C nanofibrils in 90% methanol (MeOH) solution for 5 min after removal of 1 M CPB (pH 5). The dash line denotes the peak at 1654 cm^{-1} . **c**, Comparison of N-R6-C, N-R12-C and N-R12-C(xA) in three

sections along the channel. Peak features were specifically evaluated within the amide I region. **d**, Raman results of N-R12-C fiber under various negative pressures. The solid line denotes the peak at 1664 cm^{-1} . **e**, β -sheet contents of N-R12-C fiber assembled inside the microfluidic device under various negative pressures. Peak fitting was carried out to calculate the β -sheet contents after deconvolution within the amide I region. **f**, Comparison of Raman signals for N-R6-C, N-R12-C and N-R12-C(xA) fibers formed by biomimetic spinning and manual drawing.

Comment 9: Results, line 290-294. Include references for the assignment of conformations in the amide I region.

Response: Thank you for the suggestion. Two references [31] and [32] have been added accordingly.

In Page 27, Line 808-813: “31 Lefevre, T., Paquet-Mercier, F., Rioux-Dubé, J. F. & Pérolet, M. Structure 799 of silk by Raman spectromicroscopy: From the spinning glands to the fibers. *Biopolymers* **97**, 322-336 (2012). 32 Lee, S.-M. et al. In situ Raman spectroscopic study of al-infiltrated spider dragline silk under tensile deformation. *ACS applied materials & interfaces* **6**, 16827–16834 (2014).”

Comment 10: Results, WAXD: Explain how the results were evaluated and include a reference for the theory behind it.

Response: As suggested, more discussion has been added to the revised manuscript with a new reference.

In Page 14, Line 393-398: “After normalization of peak (020),³⁵ the intensity at peak (010) suggested that the relative crystallinity of silk fibers prepared by manual drawing was positively related with the number of polyalanine blocks within the repetitive domain. In the absence of polyalanine blocks, the crystallinity of N-R12-C(xA) fiber made by sufficient manual drawing was still lower than that of N-R12-C fiber made by microfluidic spinning.”

Comment 11: Methods: Include a reference for how expression and purification were performed, and give a brief overview also here (main methods and conditions).

Response: As suggested, a reference was added together with a description of protein preparation.

In Page 18, Line 524-542: “Expression of recombinant MaSp2 was performed as previously described¹². For protein purification, cell lysis was carried out by freeze-thawing the bacterial cell pellet that had been completely resuspended in 20 mM Tris-HCl pH 7.5, 20 mM imidazole, 0.5 M NaCl supplemented with 1% Triton X-100 (Sigma-Aldrich), 0.5 mg/ml chicken egg lysozyme (Wako), 250 units TurboNuclease (Accelagen), and cOmplete EDTA-free protease inhibitor tablet (Roche). The lysate was spun down and the supernatant fraction applied to a His-TRAP FF column (2x5ml in tandem) (Cytiva), washed with 20 mM Tris-HCl pH 7.5, 20 mM imidazole, 0.5 M NaCl, and eluted with 20 mM Tris-HCl pH 7.5, 250 mM imidazole, 0.5 M NaCl. The eluate was subjected to buffer exchange against 20 mM Tris-HCl pH 7.5, 0.15 M NaCl using a VivaSpin spin concentrator with 10 kDa MWCO (Sartorius). The poly-histidine tag was removed by overnight thrombin digestion (Sigma-Aldrich) at 4°C, followed by concentration using VivaSpin. For x-R6-C, N-R6-C, N-R12-C and N-R12-C(xA), a final round of purification was carried out through LLPS, whereby 1.0 M KPi , pH 7.0, was added until two stable liquid phases were formed, followed by centrifugation and retrieval of the high-density phase, which contained highly purified MaSp2, which was then diluted in 20 mM Tris-HCl, 0.15 M NaCl. The protein concentration was measured by absorbance at 280 nm using a NanoDrop instrument (Thermo Fisher Scientific).”

Comment 12: Methods: What do you mean by “purification by LLPS”? Precipitation with phosphate?

Response: Yes. LLPS could be developed as a promising protocol to further purify the target spidroins because of its reversible feature. More details have been added to the manuscript as follows.

In Page 19, Line 536-540: “For x-R6-C, N-R6-C, N-R12-C and N-R12-C(xA), a final round of purification was carried out through LLPS, whereby 1.0 M KPi , pH 7.0, was added until two stable liquid phases were formed, followed by centrifugation and retrieval of the high-density phase, which contained highly purified MaSp2, which was then diluted in 20 mM Tris-HCl, 0.15 M NaCl.”

Comment 13: Methods: Which protein concentrations were used?

Response: 50 mg/ml. This description has been added to the revised manuscript as follows.

In Page 19, Line 542-543: “The protein concentration of 50 mg/ml was applied for the following experiments unless otherwise indicated.”

Reviewer #3

Comment 1: The author claimed that replicating the mechanical properties of the natural spider silk is still a challenge, how about the mechanical properties of the fiber fabricated in this manuscript. The systemic test, discussion, and comparison of the mechanical properties should be provided.

Response: Thank you for this question. Replicating the mechanical properties of the natural spider silk is still a challenge as the reviewer mentioned. One of the key reasons is that the spider silk assembly mechanism remains ambiguous so that it is difficult to create a native-like condition to control the sophisticated liquid-solid phase separation and mild transition of secondary structure. Therefore, our work is focused on the study of the shear effect on the fiber formation and crystallization to advance the development of spider silk assembly mechanism. To create a biomimetic condition, the negative pressure was applied to mimic the pultrusion which was revealed previously (J. Sparkes, *Nature Communications*, 2017, 8, 594) to be significant for spider silk assembly. Because of negative pressure applied, it is hardly possible to collect the long and continuous fiber under our spinning design (see Movie 1). Thus, it is out of our current scope to collect the fibers for testing mechanical properties. It is true that studying mechanical properties of MaSp2 fibers is very interesting and actually in another ongoing project we are working on such a project to make tough composite fiber by using wet spinning method.

Comment 2: Line 80-97: Many factors will affect the structure, morphology, and mechanical property of artificial spider silk, how to give an overall consideration?

Response: Thank you for this comment. It reflects the originality of this work to integrate many factors into one system, where all the native-like triggers were involved to induce the phase separation of MaSp2 with complete domains. Within this spinning system, the structure, morphology and property of artificial spider silk can be flexibly regulated.

Comment 3: Line 113: As the author claimed that reference 22 had shown fiber formation using full-domain ADF3 spidroin through sequential introduction of phosphate ions and pH decrease in a microfluidic device, what is the difference and advantage of this manuscript compared to reference 22?

Response: In comparison to reference 22, our work shows native-like spider silk assembly, where the phenomenon of LLPS was observed to form droplets and the hierarchical structure was reconstructed to visualize the nanofibrils aligning to the length of fiber. In addition, another significant feature of this work is that the shear was quantified to show the threshold value for fiber formation and its relationship with β -sheet contents. A new Supplementary Table 1 was summarized to compare our work with others to show the difference and advantage.

Supplementary Table 1. Comparison of difference on the microfluidic spinning of recombinant spidroins

	Type of protein	Domain structure	Flow driving force	Buffer system	pH drop	Fiber formed inside channels	LLPS	Hierarchical structure	Quantification of shear
This work	MaSp2	N-R6-C; N-R12-C	Negative pressure	Citrate phosphate buffer	from 7 to 5	Yes	Yes	Yes	Yes
Rammensee et al ⁵	eADF3, eADF4	N-R12-C; N-R'16-C	Syringe pump	Potassium phosphate buffer	from 8 to 6	Yes	ND	No	No
Saric et al ⁷	eADF3, eADF4	N-R'16-C; N-R12-C	Syringe pump	Potassium phosphate buffer	from 8 to 6	No	ND	No	No
Saric et al ¹⁴	TIO spidroins; eADF3; eADF4	N-R6-R'8-C; N-R12-C; N-R'16-C	Syringe pump	Potassium phosphate buffer	from 8 to 6	No	ND	No	No
Renberg et al ⁶	MaSp1	R4-C; N-R4-C	Positive pressure	Oil	No	No	No	No	No
Chen et al ¹⁵	eTuSp1	R1	Syringe pump	Isopropanol solution	No	No	No	No	No
Peng et al ¹⁶	MaSp2	R16	Syringe pump	Ethanol solution	No	No	No	No	No

ND: Not determined/discussed

Comment 4: Line 117-119: The description “quantitative analyses of the forces generated within the microfluidic channels during sample flow, and their influence on the conformation of the silk proteins, have as a rule not been carried out in rigorous detail” is incorrect, the relevant studies have been reported, for example: Adv. Mater. Technol. 2021, 2100124; ACS Sustain. Chem. Eng., 2019, 7, 14765-14774. Hence, the author should be discussed carefully the advantage of this manuscript.

Response: Thank you for the comment. These two works are very interesting and have been carefully studied in comparison to our work. There are several differences:

a) Proteins. Regenerated silk fibroin (RSF) was used in these two works and their structure and property are quite different to that of recombinant spidroins (MaSp2) employed in our study.

b) Microfluidic design. The geometry and dimensions of microfluidic chips are different. A series of native-like chemical triggers were used to induce the silk fiber assembly in our study, whereas no such triggers were applied in the case of RSF.

c) Simulation. In those previous studies there is only one phase of protein solution used in the simulation with positive pressure, whereas ours is more complicated to involve the interaction between proteins and buffers under negative pressure.

We added the discussion on these two works and then show our differences to highlight the originality of this study. The discussion in the revised manuscript is as follows:

In Page 4, Line 119-129: “Interestingly, for spinning of regenerated silk fibroin (RSF), the forces generated inside the microfluidic chip have been quantified to show the distribution of shear and elongation rate along the channel, revealing the relationship between these forces and β -sheet contents^{26,27}. However, in the case of recombinant spidroins, such quantitative analysis of the forces within microfluidic channels have not been carried out in rigorous detail to link the forces with the fiber formation and change of conformation. In comparison with RSF, recombinant spidroins are smarter in response to naturally occurring triggers with self-assembly behavior. Therefore, to quantify the forces, such as shear stress, under native-like conditions is of great interest and significance for better understanding of spider silk self-assembly mechanism.”

Comment 5: Line 154: Why did the author design section C with the same width (80 μm)?

Response: In the native spider silk gland, the convergent geometry was observed. Inspired by this natural geometry, biomimetic microfluidic chips were designed with gradually narrowed channel toward the outlet for the fabrication of silk fibers. Some representative works (J. Luo, *Int. J. Biol. Macromol.* 2014, 66, 319-324; Q. F. Peng, *Sci. Rep.* 2016, 6, 36473; L. Lu, *ACS Sustainable Chem. Eng.* 2019, 7, 14765-14774) can be found in Prof. Yaopeng Zhang’s group to show the detailed design. However, in this work, the purpose in increasing the channel width from 60 μm in section B to 80 μm in section C is to largely minimize the extensional flow so that the shear force can be highlighted to show its effect on the fiber formation and crystallization. More discussion has been added to the revised manuscript as follows.

In Page 5, Line 161-167: “It is important to state that we chose a simplified design where each individual channel has a uniform cross-section, and an increase in width from section B (60

μm) to C (80 μm), in order to focus on the effects of shear in modulating fiber assembly and crystallization with minimized effects of extensional flow. This is contrast to the more complex (i.e., convergent) geometries found in the natural spinning ducts, where the duct is progressively narrowed down toward the spinneret to enable a strong extensional flow for the alignment of silk molecules for better orientation.”

Comment 6-1: Line 232, Fig. 2: The hierarchical structure is not clear as shown in Fig. 2d. In addition, there is a big saltation of the shear stress at section B, how to affect the morphology, structure and property of the fiber.

Response: Thank you for this question. In Fig. 2d, the nanofibrils were stacked closely with each other so that it is not easy to see details of the hierarchical structure. Even though new experiments (see the Figure below) have been carried out but the observed structure of silk fiber inside the microfluidic channel is still similar to the results shown in Fig. 2d. Therefore, another experiment was designed to recover the silk fibers from the microfluidic channel and then exposing them to the Milli-Q water. Interestingly, the split nanofibrils can be clearly visualized to support hierarchical organization. Please check the hierarchical structure in the new Fig. 2f and 2g. The related discussion has been added to the revised manuscript as follows.

In Page 8, Line 254-259: “After formation, the fibers could be recovered from the device for further analysis. Significantly, the fibers were able to retain structural integrity in pure water and demonstrate high flexibility (Fig. 2f). In some localized areas, the overall fiber architecture did break apart, which clearly revealed an underlying substructure consisting of individual protein fibrils with high aspect ratio and oriented along the longitudinal axis of the fiber Fig. 2g.”

Figure. Similar results obtained to show the densely packed structure of silk fiber inside the microfluidic channel. Scale bar: 10 μm .

Fig. 2: **f**, Oriented nanofibrils observed from the flexible continuous fiber in Milli-Q water after recovering it from the microfluidic chip. Scale bar, 10 μm . **g**, Nanofibrils split from the fiber to show hierarchical organization upon exposure to the Milli-Q water. Scale bar, 10 μm .

Comment 6-2: In addition, there is a big saltation of the shear stress at section B, how to affect the morphology, structure and property of the fiber.

Response: For the fibrillation stage in section B, several triggers were put together to cause the change in morphology, structure and properties of silk fibers. More discussion has been added to the revised manuscript as follows.

In Page 8, Line 247-249: “In addition to the visible change of morphology in section B, the corresponding structure and properties of spidroins were also affected by both chemical and physical triggers introduced in this fibrillation stage.”

In Page 12, Line 351-360: “It is worth noting that while the N-R12-C was exposed to slightly higher shear in section A compared to B (Fig. 4c and e), the more considerable increase in β -sheet content (Supplementary Table 2) was detected in the latter section. It can thus be deduced that simple combination of shear and LLPS initiated by ions exchange in section A is not enough to induce the conformational transition. Analogous to the natural spider silk gland, a native-like aqueous condition was created in this work to observe relevant biomimetic chemical and physical triggers functioned synergistically in section B, allowing the emergence of β -sheets on the MaSp2 condensate material. Following section B, an enhanced shear developed the MaSp2 intermediate into solid fiber in section C accompanied by the induction of more β -sheets.”

Comment 7: Line 308-310: The quantitative analysis of β -sheet content should be provided.

Response: As suggested, quantitative analysis of β -sheet content has been added to the revised manuscript as follows.

In Page 13, Line 367-373: “Interestingly, -60 kPa was confirmed to be the threshold pressure for the formation of silk fiber, which obtained an averaged β -sheet content of 17.8%. With regard to the increase of pressure, the β -sheet content was gradually increased by 20.2% (at -70 kPa), 23.7% (at -80 kPa) and 28.9% (at -90 kPa). It was assumed that more β -sheets could be induced within silk fibers if the pressure reached beyond -90 kPa that was limited by our experimental system.”

Comment 8: Line 317: How to regulate the pH gradient of the manual drawing method. The detailed information should be added.

Response: In this work, we are not able to regulate the pH gradient by using the manual drawing method. For this reason, it shows the advantage by using the microfluidic chip which is featured capable of creating pH gradient. To avoid misunderstanding, more information has been added to the revised manuscript as follows.

In Page 13, Line 381-384: “Although pH gradient and shear cannot be induced by using the manual drawing method in comparison to the microfluidic spinning method, but the elongation force generated from manual drawing is generally considered strong and responsible for sufficient induction of β -sheets³⁴.”

Comment 9: Line 334-335: During the simulation, only Newtonian laminar flow is considered. However, there are laminar and turbulent flows in the microfluidic device. Hence, the reliability of this simulation result is not high.

Response: Thank you for this comment. To address this concern, a steady-state shear rheology was carried out using Anton Paar MCR 502 Rheometer (New Supplementary Fig. 12) to obtain the viscoelastic properties of protein solution. According to the results shown in the new Fig. 4a and 4b, a non-Newtonian behavior was confirmed for the protein solution. Therefore, the simulation was improved by using the non-Newtonian fluids to show the distribution of various ions (citrate, phosphate, chloride and sodium ions), protein concentration, shear and elongational rate/stress along the channel. The related results have been made into new Fig. 4 and discussed in the revised manuscript as follows.

In Page 14, Line 405-432: “Before modelling, the rheological experiments were performed to probe the viscoelastic properties of N-R12-C solution under concentrations from 5 to 150 mg/ml (Fig. 4a and b). In line with the results reported for other recombinant (NT2RepCT) and native spidroins,^{36,37} N-R12-C solution exhibited a typical shear-thinning behavior to see the decreased viscosity with regard to the increase of shear rate. Given a shear rate of 1 s^{-1} at $25 \text{ }^\circ\text{C}$, the shear viscosity of native spidroins was in the order of $10^3 \text{ Pa}\cdot\text{s}$, much higher than that of N-R12-C (all below $10 \text{ Pa}\cdot\text{s}$) and NT2RepCT (below $1 \text{ Pa}\cdot\text{s}$). This significant difference can be interpreted by the much higher concentration and molecular weight of native spidroins in comparison to these of recombinant spidroins. To directly compare the shear viscosity between recombinant spidroins under the same concentration (such as 100 mg/ml), the N-R12-C was remarkably higher in shear viscosity ($\sim 2 \text{ Pa}\cdot\text{s}$), more than 20 times higher than that of NT2RepCT ($0.05\text{-}0.1 \text{ Pa}\cdot\text{s}$) due to different numbers of repeat unit (12 repeats vs 2 repeats). In this work, the alanine residues in the repetitive region were found to contribute significantly to the viscosity of spidroins as evidenced by the lower viscosity (apparent to the naked eyes) of N-R12-C(xA) when compared with N-R12-C.

The shear-thinning mechanism is not fully understood, and two theories were proposed based on the colloid and polymer systems^{38,39}. In the colloid system, the phase separation during flow caused the shear thinning behavior, which was considered as phenomenon due to the disentanglement of molecular chains in the polymer system. As N-R12-C is a biopolymer and can be sheared to induce phase separation, thus its shear thinning behavior complies well with both theories. Using the Carreau model to fit the viscosity-shear rate curves under various concentrations, the non-Newtonian 3D model was well established. In this way, the distribution of various ions (citrate, phosphate, chloride and sodium ions), protein concentration, shear and elongational rate/stress along the channel were modeled (Supplementary Figs. 7-9).”

Anton Paar MCR 502 Rheometer

Supplementary Fig. 12: Experimental setup for the MaSp2 (N-R12-C) solution by using Anton Paar MCR 502 Rheometer. The tissue was wetted by the water and placed around the plate without contacting the sample. Finally, a cover was used to maintain saturated humidity of the sample.

Fig. 4: Numerical simulation of shear stress across the length of microfluidic channel. a, The viscosity-shear rate relationship of N-R12-C solution at various concentrations from 5 to 150 mg/ml. **b,** Shear-thinning behavior of N-R12-C solution at a concentration of 50 mg/ml with the result fitted by the Carreau model. **c,** Simulated shear stress distributed along the channel to show the difference in each section with the top (left) and cross-section (right) view. **d,** Comparison of shear stress s in section C when changing the negative pressure from -60 to -90 kPa. **e,** The value of shear stress estimated in sections A, B and C under different negative pressures. **f,** The β -sheet contents of N-R12-C fiber under specific shear stress estimated from the corresponding negative pressure. The red line shows the fitting result by using the quadratic function.

Comment 10: Line 393, Fig. 5: What is the meaning of the hierarchical structure, and how to distinguish this structure? What does the scale bar represent?

Response: Thank you for the reminder, the missing information for the scale bar has been added to the revised manuscript. More experiments were conducted to show a clear native-like hierarchical structure of silk fibers. Briefly, the freshly formed fibers were recovered from the device and placed in a hypotonic environment (Milli-Q water), which caused the fibers to swell. While for the most part, the fibers retained structural integrity in water, there were localized areas where the continuity of the fiber was broken, and where the underlying structures were revealed. Here, we clearly see that the fibers are composed of fine, high-aspect ratio fibrils aligned along the fiber axis. Please check the hierarchical structure in the new Fig. 2f and 2g.

The related discussion has been added to the revised manuscript as follows.

In Page 8, Line 254-259: “After formation, the fibers could be recovered from the device for further analysis. Significantly, the fibers were able to retain structural integrity in pure water and demonstrate high flexibility (Fig. 2f). In some localized areas, the overall fiber architecture did break apart, which clearly revealed an underlying substructure consisting of individual protein fibrils with high aspect ratio and oriented along the longitudinal axis of the fiber Fig. 2g.”

Fig. 2: **f**, Oriented nanofibrils observed from the flexible continuous fiber in Milli-Q water after recovering it from the microfluidic chip. Scale bar, 10 μm . **g**, Nanofibrils split from the fiber to show hierarchical organization upon exposure to the Milli-Q water. Scale bar, 10 μm .

Reviewer #4

Thank you for your valuable time in reviewing this manuscript.

REVIEWERS' COMMENTS

Reviewer #2 (Remarks to the Author):

Thanks for revising the manuscript to make the approach of the setup and study, and the results clearly presented. All points raised have been answered and incorporated in the revised manuscript. The points raised by the other referees were also handled, which improved the manuscript. I now consider the manuscript suitable for publication.

Reviewer #3 (Remarks to the Author):

As most of the comments were addressed, I recommend publishing the manuscript in the present form.